# Genetic Profiles and Risk Stratification in Adult De Novo Acute Myeloid Leukaemia in Relation to Age, Gender, and Ethnicity: A Study from Malaysia

**DOI:** 10.3390/ijms23010258

**Published:** 2021-12-27

**Authors:** Angeli Ambayya, Anthony V. Moorman, Jameela Sathar, Jeyanthy Eswaran, Sarina Sulong, Rosline Hassan

**Affiliations:** 1School of Medical Sciences, Health Campus, Universiti Sains Malaysia, Kubang Kerian 15159, Kelantan, Malaysia; angeli_100182@yahoo.com (A.A.); ssarina@usm.my (S.S.); 2Clinical Haematology Referral Laboratory, Haematology Department, Hospital Ampang, Ampang 68000, Selangor, Malaysia; jsathar@hotmail.com; 3Translational and Clinical Research Institute, Newcastle University, Newcastle upon Tyne NE2 4HH, UK; anthony.moorman@newcastle.ac.uk (A.V.M.); jeyanthy.eswaran@newcastle.edu.my (J.E.); 4Newcastle University Medicine Malaysia, Iskandar Puteri 79200, Johor, Malaysia

**Keywords:** cytogenetic, karyotype, acute myeloid leukaemia, genetics, ELN 2017

## Abstract

Hitherto, no data describing the heterogeneity of genetic profiles and risk stratifications of adult acute myeloid leukaemia (AML) in Southeast Asia are reported. This study assessed genetic profiles, Moorman’s hierarchical classification, and ELN 2017-based risk stratifications in relation to age, gender, and ethnicity in Malaysian adult AML patients. A total of 854 AML patients: male (52%), female (48%) were recruited comprising three main ethnic groups: Malays (59%), Chinese (32%) and Indians (8%). Of 307 patients with abnormal karyotypes: 36% exhibited translocations; 10% deletions and 5% trisomies. The commonest genotype was FLT3-ITD-NPM1wt (276/414; 66.7%). ELN 2017 risk stratification was performed on 494 patients, and 41% were classified as favourable, 39% as intermediate and 20% as adverse groups. More females (47%) were in the favourable risk group compared to males (37%), whereas adverse risk was higher in patients above 60 (24%) of age compared to below 60 (18%) patients. We observed heterogeneity in the distribution of genetic profiles and risk stratifications between the age groups and gender, but not among the ethnic groups. Our study elucidated the diversity of adult AML genetic profiles between Southeast Asians and other regions worldwide.

## 1. Introduction

Diagnostic cytogenetic profiles, also known as karyotypes, have been the most important predictors of outcomes in acute myeloid leukaemia over the last three decades. As proposed by Martens et al. in 2010 [1], cytogenetic aberrations are identifiable in 50% to 60% of newly diagnosed AML patients, this tool still serves as a powerful prognostic indicator for AML. Karyotypic analysis in younger patients allows prognostication of AML patients into favourable, intermediate, and adverse risk groups as recommended by the European LeukemiaNet (ELN) and the UK Medical Research Council (MRC) [2,3]. The major translocations of t(15; 17) (q22; q21), t(8; 21) (q22; q22) and inv(16) (p13q22) confer a good prognosis, whereas the loss of 5q, 7q, and complex karyotypes are associated with a dismal prognosis, leading to a higher risk of induction failure and relapse. Other chromosomal aberrations, including trisomies and normal karyotypes, are categorised as an intermediate-risk group [4,5,6,7].

The incidence of AML increases with age, with a median age of 68 years, although cases are reported at all ages [8]. In older patients. AML is characterised by profound biologic divergence, including the distribution, but not the spectrum of chromosomal aberrations [9]. Notably, favourable risk aberrations are relatively uncommon in older patients, in contrast to normal and complex karyotypes. Older patients are commonly known to have a poorer prognosis and present with high-risk cytogenetic abnormalities and preceding myelodysplastic phase or secondary leukaemia [10]. Furthermore, it is axiomatic that dismal prognosis for the older patients is typical as they also present with other comorbidities at diagnosis and express multidrug resistance with lower response to chemotherapy [9].

In recent years, the advent of molecular genetics markers, including mutations, has heralded new prognostication in AML, and today, genetic abnormalities are paramount features of AML based on the 2016 World Health Organisation classification [11]. However, relatively few data on genetics and age association have been published in the South Asia region, though studies in other geographical regions revealed that the increase in adverse cytogenetic and genetics aberrations occur continuously over a life span [11,12,13,14,15,16,17,18]. Genetic abnormalities that are essential outcome predictors, and frequently tested in AML, include FMS-like tyrosine kinase-3 gene-internal tandem (FL3-ITD) duplication and nucleophosmin-1 (NPM1) mutations recommended in clinical practice guidelines. Studies have shown that FLT3-ITD^neg^/NPM1^mut^ confers good overall survival (OS) and leukaemia-free survival (LFS) compared to FLT3-ITD^pos^/NPM1^wt^, which are known to be poor prognostic indicators, with shorter remission duration and higher relapse rates compared to FLT3-ITD^neg^/NPM1^wt^ and FLT3-ITD^neg^/NPM1^mut^ patients [19].

In Malaysia, leukaemia comprises 3.7% of the total reported cancer cases between 2012 and Myeloid leukaemia was the highest reported leukaemia in Malaysia, which displayed an increase of incidence rates with age in both males and females (1279 cases in males and 1080 cases in females; 2012–2016) (Malaysia National Cancer Registry Report 2012–2016) [20]. The association study between cytogenetic groups and the age of AML patients was firstly reported by Chin et al. in 2013 [15]. In their study, patients ranged between four months to 81 years (median, 39 years) were included, and chromosomal aberrations were reported in 30.4% of the patients while 69.6% of the patients had a normal karyotype [15]. However, little is known about the extent of cytogenetic and molecular genetics (FLT3-ITD and NPM1 mutations) association and variations based on patient age in the multi-ethnic population of Malaysia. In this study, we assessed the cytogenetics profiles, ELN based risk stratification and genetic mutations (FLT3-ITD and NPM1) with the age groups, gender, and ethnicity. The rationale behind this study was that no comprehensive studies on geographical diversity in cytogenetic abnormalities, ELN based risk stratifications, and genetic mutations (FLT3-ITD and NPM1) comprising multi-ethnic populations in adult AML had been conducted in the Southeast Asia region.

In this study, we adopted the hierarchical classification by Moorman et al. (2001) based on cytogenetic findings to study the distribution of AML based on subtypes of karyotypes compared to age among Malaysian patients [21]. The hierarchy was based on the type of chromosomal abnormality seen, but not specifically specified, classified into one of four main groups (translocation/deletion/trisomy/normal), with no consideration given to prognostication differences for different abnormalities and underlying molecular aberrations. This classification resolved the issues faced by other studies, in which patients with miscellaneous chromosome abnormalities were excluded from the study due to classification problems and analysis difficulties. We adopted this method in our study, as the rationale for the differences in the mechanisms in which these abnormalities arise [21]. Next, risk stratification by genetics, as suggested by European Leukaemia Network 2017 was also performed, in which patients were grouped into three main risk categories: favourable, intermediate, and adverse [22]. Additionally, we assessed the frequency of FLT3-ITD and NPM1 mutations in the decade-based age classification to examine the relationship between age and these mutations closely [21].

## 2. Results 

A total of 854 AML patients, including 443 male and 411 female patients, were included in this study. The median age of patients at presentation was 45 years (mean 45 years, range 12–93 years). Ethnically, Malays comprised 504 (59%) of the patients, followed by 280 (32%) Chinese and 70 (8%) Indians. Details on gender, Moorman’s classification, 2017 ELN stratification, and age by decade distribution among the ethnic group are summarised in Table 1. The distribution of cytogenetic profiles based on gender, age, group, and ethnicity are depicted in Figure 1, Figure 2 and Figure 3.

Cytogenetics were successfully performed in 90.9% (601/661) of the cases, as outlined in Table 2. Using Moorman’s classification [21], 294 (49%) of AML patients had a normal karyotype, 214 (36%) had a translocation, 60 (10%) had a deletion, and 33 (5%) had a trisomy [16]. In terms of the gender distribution of AML cases, there were slightly more males than females with a sex ratio (M: F) of 1.08 overall and 1.07 among the 601 cases with successful cytogenetics in our cohort. The Moorman clustering of four main groups (normal, translocation, deletion, and trisomy) revealed differences in terms of age distribution (χ2 = 40.14, *p* = 0.0004), as shown in Table 3. The Moorman classifications of four main groups and age groups are depicted in Figure 4. Three groups (13–15 years, 16–19 years, and ≥70 years) were excluded from the graph and all subsequent statistical analyses because the data were sparse. There was no difference in terms of ethnicity using Moorman’s classification for four main groups (χ2 = 6.80, *p* = 0.34). Subsequent analysis for the subgroups of translocations, deletions, trisomies, and normal karyotype compared to the ethnicity revealed no statistically significant association (χ2 = 24.073, *p* = 0.153). Hence, for the age-based distribution analysis, all ethnic groups were combined. 

According to the interrogation of distribution of 252 patients in this study, who had one of the eleven major chromosomal abnormalities, 41(16%) of the patients had two or more chromosomal abnormalities, as depicted in Table 4. Although only 11 abnormalities were considered, the extent and type of overlap seen are presented. By adopting Moorman’s classification [21], these 55 patients were assigned, which otherwise would have been excluded in a classification limited to major chromosomal aberrations.

In general, the frequency of AML patients increased with age. The age distribution for the translocation group of t(15:17), t(8; 21) and inv(16) declined steadily with age, whereas the cases with trisomy showed an increasing trend with age in this study. Almost half of the deletion karyotype consists of del(5q/7q) that increased in patients above 50 years of age. Similarly, about 54% of the trisomy comprised trisomy 8, increasing in patients aged 50 and above. Translocations of chromosomes were predominantly seen among the younger age groups of below In this cohort, t(15; 17) were the highest cases, with about 47% of the cases seen among young adults below 40 years of age. A similar observation was also seen among the other frequently seen t(8; 21) and inv(16), in which about 44% and 57% cases were in patients aged below 40, respectively. 

The mutational status of FLT3-ITD was available in 485/854 (57%), while NPM1 mutation findings were available in 417/854 (49%) of the cases in this cohort, as illustrated in Table 5. The age distribution of these mutations in the AML patients is shown in Figure 5. The distribution of NPM1/FLT3-ITD genotypes associated with age distribution was studied in 414 patients who had complete results for both tests. The genotype information of the 414 patients were 31(7.5%) FLT3-ITD^+^NPM1^+^, 59(14.3%) FLT3-ITD^−^NPM1^+^, 48(11.6%) FLT3-ITD^+^NPM1^wt^, 276(66.7%) FLT3-ITD^−^NPM1^wt^. (Table 5, Figure 6) The genotype, FLT3-ITD^+^NPM1^+,^ was present with higher percentages (13%) in the 40–49 years and 50–59 years groups. Secondly, FLT3-ITD^−^NPM1^+^ genotypes were seen predominantly in two age groups, 40–49 (21%) years and 60–69 years (20%) old AML patients. The frequency of FLT3-ITD^−^NPM1^wt^ was significantly higher in the older than 50 age groups, whereas the FLT3-ITD^+^NPM1^+^ fraction decreased significantly in the patients above 60 years of age. The FLT3-ITD^−^NPM1^+^ and FLT3-ITD^+^NPM1^wt^ patients displayed fluctuations at different age groups with a notable decrease in the frequencies between 50 and 59 years of age. Comparing patients below 50 years to patients above 50 years, no statistically significant difference (χ2 = 1.063, *p* > 0.05) in the occurrence of these NPM1/FLT3-ITD subgroups was observed. In terms of ethnicity, NPM1 mutations were lower among Chinese (14%) compared to Malays (24%) and Indians (29%) (χ2 = 6.726, *p* = 0.034). However, FLT3-ITD mutations and genotypes (NPM1/FLT3-ITD) did not reveal any significant association between the ethnic groups in this study (χ2 = 10.279, *p* = 0.113).

A total of 494 patients were stratified based on 2017 ELN’s recommendation based on the availability of cytogenetics and molecular results (FLT3-ITD and NPM1 mutations), of which 201 (41%) were grouped as favourable, 195 (39%) as intermediate and 98 (20%) as adverse. The median age for this analysis was 42 years, ranging between 15 and 89 years. There was a significant difference in risk group distribution among the below and the above 60 age groups (χ2 = 12.554, *p* = 0.0019). Among the below 60 age group, 44% were assigned into the favourable risk group, and 18% of the patients were grouped as the adverse risk group. However, in the above 60 groups, only 24% of the patients were categorised as favourable risk, whereas about 29% of the patients were assigned to the adverse risk group. Overall, more females were assigned to the favourable group (47%) compared to males (34%), and males were slightly more than the females in the adverse risk group (χ2 = 8.931, *p* = 0.01149). In terms of ethnicity comparison, the distribution of favourable risk patients was approximately even; Malays (42%), Chinese (38%), and Indians (43%), but adverse risk patients were higher among the Indians (35%) compared to Malaysia (17%) and Chinese (31%) (χ2 = 10.849, *p* = 0.02). 

## 3. Discussion

AML is a heterogeneous disorder present with a plethora of clinicopathological features, including cytogenetic and molecular genetic findings [23,24,25]. Although studies have revealed geographical diversity in cytogenetic aberrations in haematological malignancies, no comprehensive studies encompass the multi-ethnic population in Southeast Asia [12,13,14,16,18,26,27]. To the best of our knowledge, this is the largest cohort in this region that reports findings from three major ethnic groups in Malaysia, Malay, Chinese and Indians [15,26].

The median age of patients at presentation was 45 years (mean 45), with approximately 77% of the patients being below 60 years of age. Reports from some of the Asian countries disclosed the age of presentation of AML was 48 years (median) in Singapore [26], 51 years (mean) in Japan [16] and 38 years (median) in India [18] and 37.5 years (median) in Pakistan [17]. The age at presentation varied in studies from Europe, America, England, Arab and Africa, ranging between 45 to 71 years [3,11,13,14,28,29]. The age variations are not ascertained whether it is related to geographical regions or ethnicity or due to the patient referral system in the study region [26]. In our study, multi-ethnic (Malay, Chinese and Indians) AML patients were referred from other tertiary hospitals in Peninsular Malaysia as we serve as the national referral centre for adult haematology patients in Malaysia. Similar to Enjeti et al. (2004) in a study in Singapore, comprising similar ethnic groups of Malays, Chinese and Indians, no ethnicity-based cytogenetic profiles were compared as no significant differences were observed [26].

Fourteen previous studies that interrogated the association between age and chromosomal aberrations in AML exhibited corroboration with this study as depicted in Table 6. Normal karyotypes were reported with varying frequencies between different countries ranging between 25% in a study by Lazarevic et al. (2014) in Sweden and 70% in another study in Malaysia that encompasses the paediatric AML. Most studies were consistent with our findings (49%), including studies in other Asian countries that only included adult AML [13,14,18]. The higher frequencies of normal karyotype in some of these studies could be due to an admixture of undiscovered chromosomal abnormalities and cryptic aberrations beyond the detectable resolution of conventional cytogenetic cells [30,31,32,33,34,35]. The frequency of normal karyotype almost remained constant between the age of 20–49 years and raised at the age group of 50–59 years and then declined at age 60 and above. These findings were not supported by several studies [11,21,36,37,38,39], although additional support for our findings on the decrease in the frequency of normal karyotypes on the older patients was supported by some studies [40,41,42].

Cytogenetic aberrations were detected in 51% of the cases, commensurable to other cohorts from various geographical locations (range 42–75% abnormalities) [13,14,18,27,28,30,42,43]. Balanced translocations in this cohort were comparable with other studies in other geographical regions. The t(15; 17) was the commonest translocation and was reported in 14% of the AML patients, almost similar with the observation in Japan (14%), Australia (14%), China (14%), England(12%), Tunisia (13.1%), and Singapore (11%). In some other countries, t(15; 17) were reported less frequently, such as in Morocco (3.9%), Pakistan (4.9%). The higher frequencies of t(15; 17) could be due to the inclusion of children in some of these studies [3,11,18,27]. The t(8; 21) was the second most common translocations reported in our cohort, in parallel with other adult AML studies, including some countries in the Asian regions (China and Singapore) [10,11,21,26,27]. However, in some Asian countries, t(8; 21) were the most common translocations among the adult AML patients as reported by cohorts in Pakistan [17], India [18], Japan [16] as well as in Moroccan [13] AML patients. Contrary to Enjeti et al. (2004) [26], there is a lack of evidence that t(15; 17) were higher in Asians as Pakistan, Japan, and India reported higher frequencies of t(8; 21) compared to t(15; 17) [16,17,18]. No evidence was available if these lower frequencies were due to geographical location or other confounding factors such as different banding techniques and leukemic cell proliferation during culture that could result in underestimation by the conventional cytogenetic method as technical caveats are not discussed in these cohorts [12,13,16,17,18,26,27]. In our cohort, we affirm that these higher frequencies of t(15; 17) compared to the t(8; 21) are true findings and were in accordance with the molecular genetics screening using the Leukaemia Q-Fusion Screening Kit as follow: 75/341 (21.99%) were positive for t(15; 17) and 50/459 (10.89%) were positive for t(8; 21) as detailed in Appendix A.

The inv(16) were seen in about 5% of the cases, which concorded findings in most countries except Singapore, Sweden, and Pakistan that reported only 1% of cases. However, the chromosomal aberration in inv(16) is subtle can be missed in cases with compromised preparation [26]. The t(11q23) are concordant with findings from other parts of the world, comprising between 1–5% of the cases. The most frequent numerical chromosomal aberration was partial and/or complete deletions of chromosomes 5 and 7, cumulatively accounting for 5% of the cases. In contrast to Cheng Y et al. (2009) [27], there is no evidence that the frequency of these aberrations was higher in the western countries because studies in Japan, India and Singapore reported higher frequencies of del(5/7) between 9–14%.

Almost all the studies showed that the del(5q/7q) and trisomy 8 increased with age [5,11,21,36,37,38,40,41,42,44]. Findings from ten studies agreed with the frequency of translocation that decreased with age [9,11,36,38,39,40,42,44]. Nonetheless, this finding did not agree with several other studies [5,12,16,27,28,39]. These two abnormalities (deletion and translocation) possess different putative oncogenetic consequences and are likely to arise from different types of DNA damages, indicating two discrete groups [30,45]. Increase of patients with deletions concords with the number of mutations in proportion to age due to prolonged exposure to environmental carcinogens [46]. However, the age distribution of patients with a translocation may not result from a series of mutation accumulations from environmental hazards but rather an indication that other confounding non-environmental factors or pathogenesis are associated with a single rate specific mutation [47]. However, the mechanism that triggers the translocations are yet to be understood [48].

Our findings on the presence of NPM1 and FLT3-ITD mutations that decreased with age, at 50 years and above were supported by a study by Schneider et al. (2012) [49] (Appendix A). These data conflicted with previously published reports that described that NPM1 mutation was more frequently seen in the elderly patients, although their cohorts admixed different chromosomal aberrations of AML [50,51,52,53]. Similarly, FLT3-ITD mutations exhibited a substantial dependence on age in accordance with previous studies [49,54]. Similar to other published literature, a decrease of favourable and clinically relevant genotype, FLT3-ITD-NPM1+ group cannot be substantiated by our results though this genotype confers better prognosis among the elderly AML patients [49,55].

The ELN 2017 risk stratifications on this cohorts agreed with the reports that stated more females in the favourable prognosis group and more males in the adverse risk groups (χ2 = 8.931, *p* = 0.01149), especially in the below 60 AML patients [56,57]. There was no significant difference observed in the favourable risk stratification among the ethnic groups. In the adverse risk group, Indians were higher than Malays and Chinese (χ2 = 10.849, *p* = 0.02), but the number of Indian patients was too small (*n* = 49) for subset analysis. We only used several genetic markers suggested in the ELN-2017 in our resource-limited laboratory testing, so the risk stratifications may require refinements if more genetic markers were included.

In conclusion, our data elucidated the cytogenetic and genotype profiles of NPM1 and FLT3-ITD mutations in relation to age in AML patients in Malaysia. Although Chin et al. (2013) reported the cytogenetic profile of AML in Malaysia, it was done on a smaller cohort of 480 cases compared to the current study, which included 854 AML patients. The normal karyotype percentage in their cohort was 69.6%, much higher than our cohort (49%). Due to higher percentages of normal karyotypes reported in their cohort, other chromosomal abnormalities reported were much lower compared to our cohort as they have included AML patients between 0–60 years of age. We could not perform age-specific incidence of AML in our cohort as the AML patients came from different locations from Peninsular Malaysia. Although the bulk of the haematological malignancies patients from Peninsular Malaysia were referred to our centre, some patients were treated in various centres such as teaching hospitals and private medical centres, and so the incidence calculations resulted in lower incidences compared to the National Cancer Registry Report, 2012–2016 that do not specify the subtypes of AML.

To summarise, based on cytogenetic abnormalities and age group comparison, translocation and deletion karyotypes have different age distributions pattern. These findings fundamentally reiterate that the aetiology of these two types of abnormalities (deletion and translocation) may be distinct. With age, the number of normal karyotypes increases, suggesting that further genetic testing is required in patient management in terms of choice of therapies tailored to specific genetic aberrations that could improve patients’ prognostication in the future. As only limited genetic markers were tested in our routine AML diagnostic workup, the normal karyotypes in this study could have other undetected genetic aberrations. The differences in the genotype of NPM1 and FLT3-ITD mutations among the age groups shed light on the disease biology and partially explain the dismal prognosis, especially in elderly patients. The hierarchical cytogenetic classification adopted in this study, alongside the genotypes (FLT3-ITD and NPM1), ELN 2017 risk stratification, provided a groundwork for etiological studies in Malaysia. Our study is the first and largest study to compare the cytogenetic profiles, ELN 2017 based risk stratifications, NPM1/FLT3-ITD mutations associated with age among the AML patients in Malaysia and Southeast Asia.

## 4. Materials and Methods

Data of de novo AML patients diagnosed between 1 January 2012 until 30 June 2019 were collected as previously described [58,59]. These patients consist of three major ethnic groups (Malay, Chinese and Indian), aged between 13 and 93 years old, referred to Haematology Department, Hospital Ampang, which serves as the national referral centre of haematology in Malaysia. All laboratory reports (cytogenetic findings, 30 fusion genes RT-PCR results, FLT3-ITD and NPM1 mutation status) were retrieved from Clinical Haematology Referral Laboratory, Hospital Ampang. Cytogenetics analyses were performed according to standard culturing and banding techniques. Scrutinisation of successful cytogenetic analyses was based on the presence of a minimum twenty normal metaphases or if a clonal chromosomal aberration was detected with the inclusion of only stem-line clonal abnormalities. International System for Human Cytogenetic Nomenclature (2016) was used to describe clonal chromosomal aberrations [60]. Molecular genetics were performed by qualitative polymerase chain reaction for FLT3-ITD mutation detection and high-resolution melting analysis for NPM1 mutation detection.

Qualitative multiplex 2-step real-time PCR assay using Leukaemia Q-Fusion Screening Kit (QuanDx, San Jose, CA, USA) for simultaneous detection of 30 fusion genes according to standard protocols with bone marrow or peripheral blood. For cases with 10–20 normal metaphases with no chromosomal abnormality by cytogenetics, PCR assay using Leukaemia Q-Fusion Screening Kit was utilised for assignment into normal karyotype group (Appendix A). Molecular genetics results detected using Leukaemia Q-Fusion Screening Kit include BCR-ABL, PML-RARA, RUNX1-RUNX1T1, CBFB-MYH11, MLL, and FUS-ERG were used to assign patients into corresponding cytogenetics groups in cases where no karyotype was available (no results/failed karyotype/insufficient metaphase cells) and in cases with normal karyotypes (>20 cells; 10–20 cells). In addition, aberrations that were missed by cytogenetic analyses but detected using the Leukaemia Q-Fusion Screening Kit are summarised in Appendix A. Summary of mutations (FLT3-ITD/NPM1) and fusions detected using Leukaemia Q-Fusion Screening Kit is detailed in Appendix A.

As described, we adopted Moorman’s classification [16], in which all cases with successful cytogenetic analyses were divided into four mutually exclusive karyotype groups (normal, translocation, deletion and trisomy) [21]. For statistical analyses, age groups were divided into 13–15 years, 16–19 years, 20–29 years, 30–39 years, 40–49 Years, 50–59 years, 60–69 years and >70 years for the hierarchical classification [21]. The percentages (relative data) of each cytogenetic subtype and FLT3-ITD and NPM1 mutations in age groups adopted from Moorman’s classification were presented to depict age-related variations of genetic aberrations [21,28]. Based on ELN 2017 recommendations, patient’s risk stratification was determined for 494 patients with complete cytogenetic and molecular genetic findings (FLT3-ITD, NPM1 mutational status). Patients’ association between the ELN 2017 risk stratification, FLT3-ITD/NPM1 genotype and gender, age groups (below 60 and above 60 years of age) and ethnicity (Malay, Chinese, Indian) were assessed. Chi-Square analyses were used to identify differences between age cohorts with p-values less than 0.05 that were considered statistically significant.

## Figures and Tables

**Figure 1 ijms-23-00258-f001:**
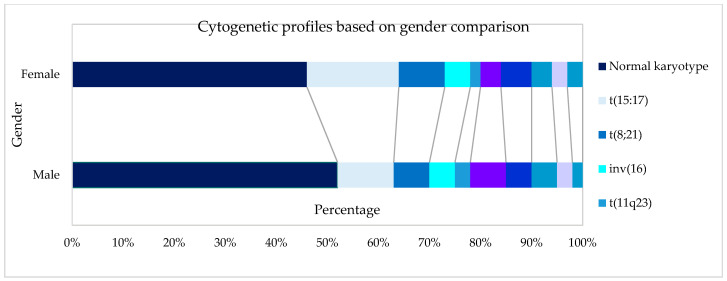
Cytogenetic profiles based on gender.

**Figure 2 ijms-23-00258-f002:**
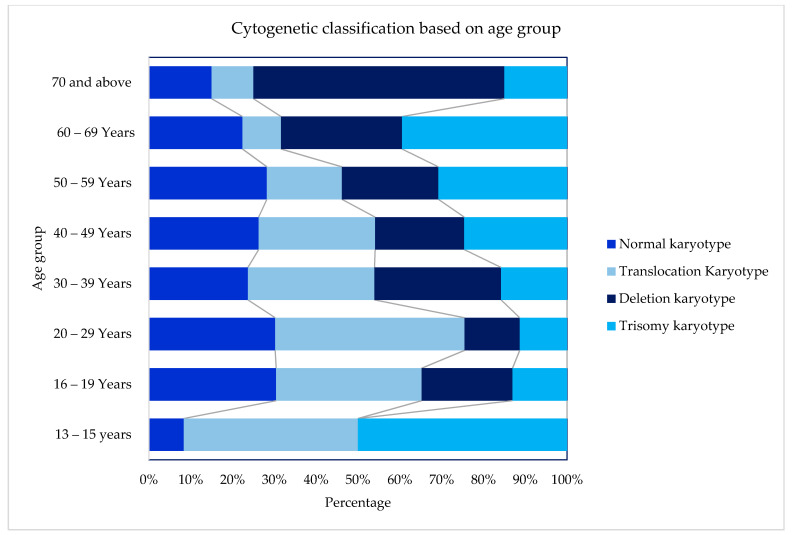
Cytogenetic profiles based on age groups.

**Figure 3 ijms-23-00258-f003:**
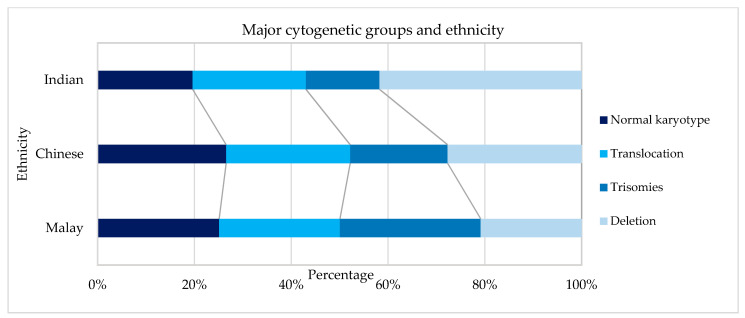
Cytogenetic profiles based on ethnicity.

**Figure 4 ijms-23-00258-f004:**
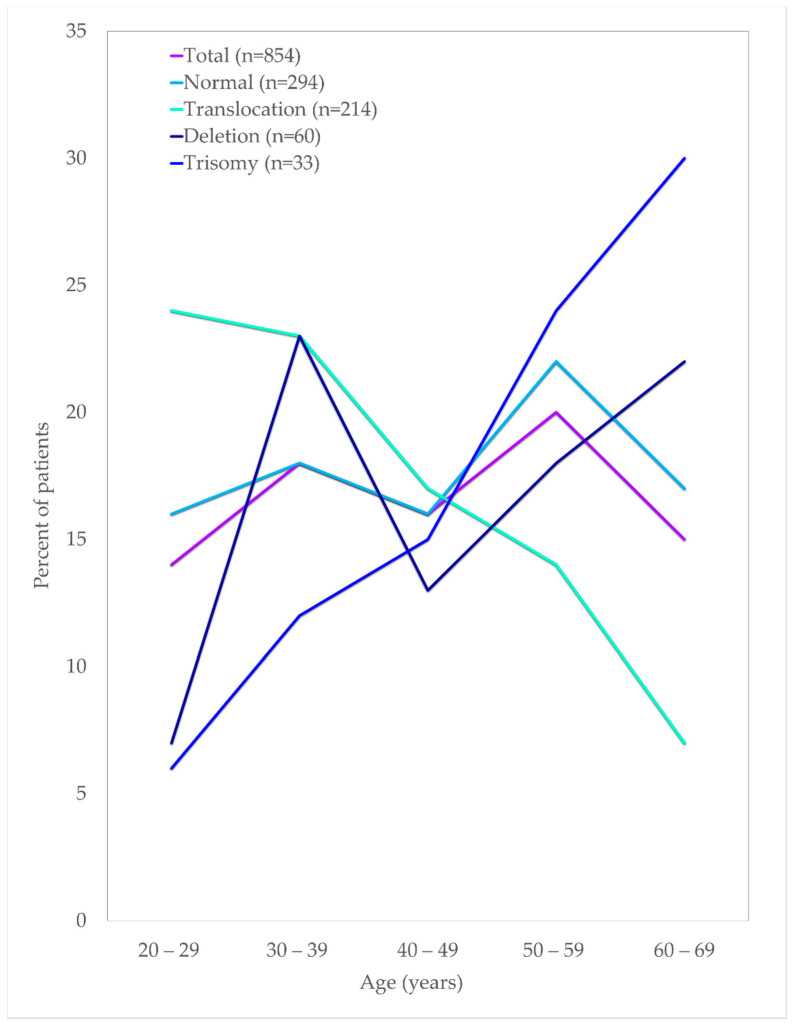
Age distribution of AML patients by their cytogenetic findings.

**Figure 5 ijms-23-00258-f005:**
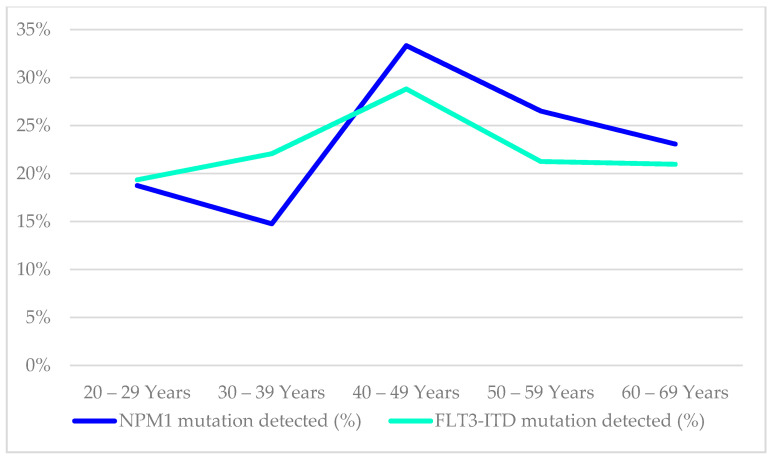
Age distribution of AML patients by their NPM1 and FLT3-ITD mutational findings.

**Figure 6 ijms-23-00258-f006:**
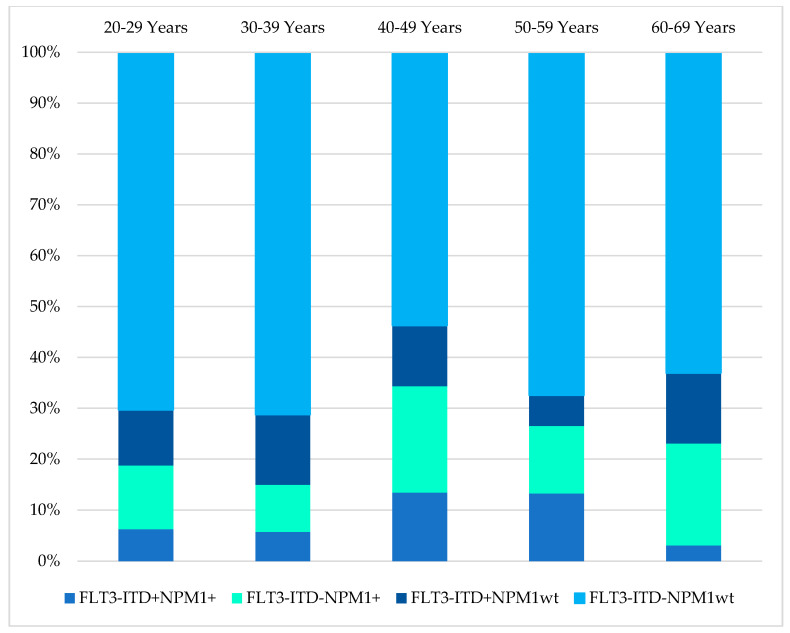
Age distributions of AML patients by their FLT3-ITD/NPM1 mutation status.

**Table 1 ijms-23-00258-t001:** Summary on gender, Moorman’s classification, 2017 ELN stratification, and age by decade distribution among the ethnic groups (number (%)).

Factors	Male	Female	*p*-Value ^¥^	Malay	Chinese	Indian	*p*-Value ^£^	Total
**Age (years)**								**854 (100)**
13–15 years	16 (2)	8 (1)		19 (4)	5 (2)	0 (0)		24 (3)
16–19 Years	29 (3)	29 (3)		42 (8)	10 (4)	6 (9)		58 (6)
20–29 Years	66 (8)	58 (7)		84 (17)	27 (10)	13 (19)		124 (15)
30–39 Years	73 (9)	79 (9)		94 (19)	44 (16)	14 (20)		152 (18)
40–49 Years	62 (7)	75 (9)		83 (16)	46 (16)	8 (11)		137 (16)
50–59 Years	89 (10)	79 (9)		94 (19)	59 (21)	15 (21)		168 (19)
60–69 Years	71 (8)	57 (7)		69 (14)	52 (19)	7 (10)		128 (15)
70 and above	37 (4)	26 (3)		19 (4)	37 (13)	7 (10)		63 (7)
**Below 60 and above 60**								**854 (100)**
<60 years	335 (39)	328 (38)	0.140	416 (49)	191 (22)	56 (7)	0.000	663 (77)
>60 years	108 (13)	83 (10)		88 (10)	89 (10)	14 (2)		191 (23)
**Ethnicity**								**854 (100)**
Malay	257 (30)	247 (29)						504 (59)
Chinese	149 (17)	131 (15)						280 (32)
Indian	37 (4)	33 (4)						70 (8)
**White blood cell count (109/L) ^a^**								**740 (100)**
<100 (10^9^/L)	323 (85)	301 (84)	0.600					624 (84)
>100 (10^9^/L)	57 (15)	59 (16)						116 (16)
**Karyotype**								**854 (100)**
Normal karyotype	161 (52)	133 (46)	0.720	177 (49)	94 (50)	23 (42)		294 (49)
Translocation Karyotype								
t(15:17)	34 (11)	52 (18)		53 (15)	26 (14)	7 (13)		86 (14)
t(8; 21)	23 (7)	28 (10)		30 (8)	19 (10)	2 (4)		51 (8)
inv(16)	16 (5)	14 (5)		14 (4)	11 (6)	5 (9)		30 (5)
t(11q23)	9 (3)	6 (2)		12 (3)	1 (1)	2 (4)		15 (2)
Miscellaneous translocations	21 (7)	11 (4)		19 (5)	9 (5)	4 (7)		32 (5)
Deletion karyotype								
del(5q/7q)	16 (5)	16 (5)		12 (3)	14 (7)	6 (11)		32 (5)
Miscellaneous deletions	15 (5)	13 (4)		18 (5)	6 (3)	4 (7)		28 (5)
Trisomy karyotype								
+8	9 (3)	10 (3)		11 (3)	6 (3)	2 (4)		19 (3)
Miscellaneous trisomies	6 (2)	8 (3)		12 (3)	2 (1)	0 (0)		14 (2)
Total	0	0		0	0	0		0
**Moorman’s Classification (% of successful cases)**								**854 (100)**
Normal karyotype	161 (52)	133 (46)	0.720	177 (49)	94 (50)	23 (42)	0.3400	294 (49)
Translocation Karyotype	103 (33)	111 (38)	0.810	128 (36)	66 (35)	20 (36)		214 (36)
Deletion karyotype	31 (10)	29 (10)	0.890	30 (8)	20 (11)	10 (18)		60 (10)
Trisomy karyotype	15 (5)	18 (6)	0.120	23 (6)	8 (4)	2 (4)		33 (5)
**Gene mutations**								
NPM1 ^b^								**419 (100)**
Detected	41 (19)	49 (25)	0.16	61 (24)	18 (14)	11 (29)	0.030	90 (22)
Not Detected	176 (81)	151 (75)		191 (76)	109 (86)	27 (71)		327 (78)
FLT3-ITD ^c^								**485 (100)**
Detected	47 (19)	37 (16)	0.42	51 (17)	24 (17)	9 (21)	0.8	84 (17)
Not Detected	205 (81)	196 (84)		247 (83)	120 (83)	34 (79)		401 (83)
Genotype (FLT3-ITD/NPM1) ^d^								**414 (100)**
FLT3-ITD^mut^ NPM1^mut^	15 (7)	16 (8)	0.37	21 (8)	8 (6)	2 (5)	0.113	31 (7)
FLT3-ITD^mut^ NPM1^wt^	29 (13)	19 (10)		27 (11)	15 (12)	6 (16)		48 (12)
FLT3-ITD^neg^NPM1^mut^	26 (12)	33 (26)		40 (16)	10 (8)	9 (24)		59 (14)
FLT3-ITD^neg^ NPM^wt^	145 (67)	129 (65)		160 (65)	93 (74)	20 (54)		276 (67)
**Cytogenetic risk based on ELN 2017 ^e^**								**494 (100)**
Favourable	85 (34)	116 (47)	0.011	121 (42)	59 (38)	21 (43)	0.020	201 (41)
Intermediate	109 (44)	86 (35)		119 (41)	65 (42)	11 (22)		195 (39)
Adverse	55 (22)	43 (18)		50 (17)	31 (20)	17 (35)		98 (20)
ELN and age groups								**494 (100)**
*Below 60 ^f^*								**408 (83)**
Favourable	77 (39)	103 (49)	0.110	109 (44)	53 (45)	18 (45)	0.02	180 (44)
Intermediate	79 (40)	76 (36)		101 (40)	46 (39)	8 (20)		155 (38)
Adverse	41 (34)	32 (15)		40 (16)	19 (29)	14 (35)		73 (18)
*Above 60 ^g^*								**86 (17)**
Favourable	8 (15)	13 (38)	0.010	12 (30)	6 (16)	3 (33)	0.00	21 (24)
Intermediate	30 (58)	10 (29)		18 (45)	19 (51)	3 (33)		40 (47)
Adverse	14 (27)	11 (32)		10 (25)	12 (32)	3 (33)		25 (29)

^¥^*p*-value < 0.05 is significant for gender comparison. ^£^
*p*-value < 0.05 is significant for ethnicity comparison. ^a^ WBC results were not available in 114 patients (63 males,51 females). ^b^ NPM1 mutation were not tested in 437 patients (226 males,211 females). ^c^ FLT3-ITD were not tested in 369 patients (191 males, 178 females). ^d^ FLT3-ITD/NPM1 genotype not tested in 440 patients (228 males, 214 females). ^e^ Incomplete information for risk stratification in 360 patients (194 males, 166 females). ^f^ Incomplete information for risk stratification in below 60 age groups in 255 patients (138 males,117 females). ^g^ Incomplete information for risk stratification in above 60 age groups in 105 patients (56 males, 49 females).

**Table 2 ijms-23-00258-t002:** Number (%) ^a^ of AML patients by gender and cytogenetic findings.

Cytogenetics Group	Total ^a^	Males	Females	Sex Ratio (M:F)
Cytogenetics Investigation				
Total cases	854 (100)	443 (100)	411 (100)	1.08
Successful	601 (70)	310 (70)	291 (71)	1.07
Failed	60 (7)	31 (7)	29 (7)	1.07
Not tested	193 (23)	102 (23)	91 (22)	1.12
Karyotype Group ^b^				
**Normal karyotype**	**294 (49)**	**161 (52)**	**133 (46)**	1.21
**Translocation Karyotype**	**214 (36)**	**103 (33)**	**111 (38)**	0.93
t(15:17)	86 (14)	34 (11)	52 (18)	0.65
t(8; 21)	51 (9)	23 (7)	28 (9)	0.82
inv(16)	30 (5)	16 (5)	14 (5)	1.14
t(11q23)	15 (3)	9 (3)	6 (2)	1.5
Miscellaneous translocations	32 (5)	21 (7)	11 (4)	1.91
**Deletion karyotype**	**60 (10)**	**31 (10)**	**29 (10)**	1.07
del(5q/7q)	32 (5)	16 (5)	16 (6)	1
Miscellaneous deletions	28 (5)	15 (5)	13 (4)	1.15
**Trisomy karyotype**	**33 (5)**	**15 (5)**	**18 (6)**	0.83
+8	19 (3)	9 (3)	10 (3)	0.9
Miscellaneous trisomies	14 (2)	6 (2)	8 (3)	0.75

^a^ some percentages may not be 100 in total due to rounding of decimal points. ^b^ Percentages of successful cases.

**Table 3 ijms-23-00258-t003:** Number (%) ^a^ of AML patients by age and cytogenetic findings.

Cytogenetic Group	Total	13–15 Years	16–19 Years	20–29 Years	30–39 Years	40–49 Years	50–59 Years	60–69 Years	≥70
**Cytogenetic investigation**									
**Total cases**	854 (100)	24 (3)	58 (7)	124 (14)	152 (18)	137 (16)	168 (20)	128 (15)	63 (7)
Successful	601 (100)	16 (3)	42 (7)	103 (17)	119 (20)	98 (16)	115 (19)	86 (14)	22 (4)
Failed	60 (100)	3 (5)	2 (3)	8 (13)	9 (15)	9 (15)	16 (27)	11 (18)	2 (3)
Not tested	193 (100)	5 (3)	14 (7)	13 (7)	24 (12)	30 (16)	37 (19)	31 (16)	39 (20)
**Karyotype Group**									
**Normal karyotype**	**294 (100)**	**3 (1)**	**21 (7)**	**46 (16)**	**52 (18)**	**48 (16)**	**65 (22)**	**49 (17)**	**10 (3)**
**Translocation Karyotype**	**214 (100)**	**11 (5)**	**17 (8)**	**51 (24)**	**49 (23)**	**37 (17)**	**31 (14)**	**14 (7)**	**4 (2)**
t(15:17)	86 (100)	2 (2)	8 (9)	20 (23)	21 (24)	17 (20)	12 (14)	5 (6)	1 (1)
t(8; 21)	51 (100)	5 (10)	8 (16)	12 (23)	11 (21)	6 (12)	6 (12)	1 (2)	2 (4)
inv(16)	30 (100)	1 (3)	1 (3)	8 (27)	9 (30)	4 (13)	7 (23)	0 (0)	0 (0)
t(11q23)	15 (100)	0 (0)	0 (0)	5 (33)	2 (13)	3 (20)	1 (7)	4 (27)	0 (0)
Miscellaneous translocations	32 (100)	3 (9)	0 (0)	6 (19)	6 (19)	7 (22)	5 (16)	4 (12)	1 (3)
**Deletion karyotype**	**60 (100)**	**0 (0)**	**3 (5)**	**4 (7)**	**14 (23)**	**8 (13)**	**11 (18)**	**13 (22)**	**7 (12)**
del(5q/7q)	32 (100)	0 (0)	2 (6)	0 (0)	6 (19)	2 (6)	9 (28)	9 (28)	4 (13)
Miscellaneous deletions	28 (100)	0 (0)	1 (4)	4 (14)	8 (29)	6 (21)	2 (7)	4 (14)	3 (11)
**Trisomy karyotype**	**33 (100)**	**2 (6)**	**1 (3)**	**2 (6)**	**4 (12)**	**5 (15)**	**8 (24)**	**10 (30)**	**1 (3)**
+8	19 (100)	2 (11)	1 (5)	1 (5)	2 (11)	1 (5)	5 (26)	6 (32)	1 (5)
Miscellaneous trisomies	14 (100)	0 (0)	0 (0)	1 (7)	2 (14)	4 (29)	3 (21)	4 (29)	0 (0)

^a^ some percentages may not be 100 in total due to the rounding of decimal points.

**Table 4 ijms-23-00258-t004:** Distribution of 252 patients with ≥1 major chromosomal aberration in AML.

Abnormality	t(15:17)	t(8; 21)	inv(16)	t(11q23)	t(3q; 21)	del(5q)	del(7q)	del(9q)	+8	+21	+22	Total Abnormalities
t(15:17)	77	0	0	0	0	0	2	0	3	4	0	86
t(8; 21)	0	47	0	0	0	0	1	2	0	1	1	52
inv(16)	0	0	25	0	0	0	1	0	0	1	3	30
t(11q23)	0	0	0	13	0	0	0	0	1	1	0	15
t(3q; 21)	0	0	0	0	2	0	1	0	0	0	0	3
del(5q)	0	0	0	0	0	4	10	2	4	1	3	24
del(7q)	2	1	1	0	1	10	10	2	5	1	2	35
del(9q)	0	2	0	0	0	2	2	8	1	1	0	16
+8	3	0	0	1	0	4	5	1	17	1	1	33
+21	4	1	1	1	0	1	1	1	1	7	1	19
+22	0	0	3	0	0	3	2	0	1	1	1	11
Total abnormalities	86	51	30	15	3	24	35	16	33	19	12	324

**Table 5 ijms-23-00258-t005:** FLT3-ITD and NPM1 mutation distribution by age ^a^.

Age Groups	NPM1 Mutation Detected (%)	Total (*n*) Tested for NPM1	FLT3-ITD Mutation Detected (%)	Total (n) Tested for FLT3-ITD	FLT3-ITD+NPM1+	FLT3-ITD-NPM1+	FLT3-ITD+NPM1^wt^	FLT3-ITD-NPM1^wt^	Total (*n*) Tested for Both FLT3-ITD and NPM1
13–15 years	0 (0)	5	0 (0)	14	0 (0)	0 (0)	0 (0)	5 (100)	5
16–19 Years	3 (12)	26	5 (16)	32	0 (0)	3 (12)	5 (19)	18 (69)	26
20–29 Years	12 (19)	64	12 (16)	74	4 (6)	8 (13)	7 (11)	45 (70)	64
30–39 Years	13 (15)	88	17 (18)	94	5 (6)	8 (9)	12 (14)	62 (71)	87
40–49 Years	23 (33)	69	17 (22)	76	9 (13)	14 (21)	8 (12)	36 (54)	67
50–59 Years	22 (27)	83	17 (18)	97	11 (13)	11 (13)	5 (6)	56 (67)	83
60–69 Years	15 (23)	65	13 (17)	75	2 (3)	13 (20)	9 (14)	41 (63)	65
70 and above	2 (12)	17	3 (13)	23	0 (0)	2 (12)	2 (12)	13 (76)	17
Total	90	417	84	485	31	59	48	276	414

^a^ The mutational status of FLT3-ITD and NPM1 were assessed as follow: presence of both FLT3-ITD (FLT3-ITD^+^NPM1^+^) and NPM1 mutations/FLT3-ITD mutation detected with NPM1 wild type (FLT3-ITD^+^NPM1^wt^)/FLT3-ITD mutation not detected with NPM1 mutation present (FLT3-ITD^−^NPM1^+^)/no FLT3-ITD or NPM1 mutation detected (FLT3-ITD^−^NPM1^wt^).

**Table 6 ijms-23-00258-t006:** Number (%) of AML patients by age and cytogenetic profiles based on published studies.

				Cytogenetic Group								
Reference Year	Country	Age Band (Years)	Total	Normal	t(15; 17)	t(8; 21)	inv(16)	t(11q23)	del(5q/7q)	del(5q)	del(7q)	Trisomy 8
Grimwade et al., 1998 [5] ^§^	UK	Total	1612 (100)	680 (42)	198 (12)	122 (8)	57 (4)	60 (4)	-	28 (2)	32 (2)	148 (9)
		0–14	340 (21)	91 (27)	31 (9)	41 (12)	16 (5)	26 (8)	-	4 (1)	7 (2)	46 (14)
		15–34	461 (29)	177 (38)	87 (19)	28 (6)	26 (6)	21 (5)	-	5 (1)	8 (2)	47 (10)
		35+	811 (50)	412 (51)	80 (10)+	53 (7)+	15 (2)+	13 (2)+	-	19 (2)	17 (2)	55 (7)
Moorman et al., 2001 [21]	UK	Total	593 (100)	242 (100)	74 (100)	34 (100)	27 (100)	12 (100)	61 (100)	-	-	37 (100)
		16–19	20 (3)	8 (3)	3 (4)	1 (3)	1 (4)	0 (0)	1 (2)	-	-	1 (3)
		20–29	77 (13)	21 (9)	17 (23)	10 (30)	7 (26)	3 (25)	1 (2)	-	-	6 (16)
		30–39	80 (13)	27 (11)	14 (19)	8 (24)	3 (11)	6 (50)	5 (8)	-	-	5 (14)
		40–49	98 (17)	40 (17)	15 (20)	5 (15)	7 (26)	2 (17)	6 (10)	-	-	5 (14)
		50–59	117 (20)	50 (21)	15 (20)	4 (12)	6 (22)	0 (0)	13 (21)	-	-	8 (22)
		60–69	201 (34)	96 (40)	10 (14)	6 (18)	3 (11)	1 (8)	35 (57)	-	-	12 (32)
Creutzig et al., 2016 [11]	Germany	Total	5564 (100)	2394	311	256	99	140				81
		0 – <2	271 (5)	37 (2)	5 (2)	1 (0)	17 (17)	68 (49)	NA	-	-	5 (6)
		2 – <12	477 (9)	96 (4)	28 (9)	82 (32)	27 (27)	27 (19)	NA	-	-	7 (8)
		12 – <18	444 (8)	121 (5)	52 (17)	59 (23)	24 (24)	20 (14)	NA	-	-	13 (16)
		18 – <40	417 (7)	182 (8)	52 (17)	33 (13)	15 (15)	14 (10)	NA	-	-	10 (12)
		40 – <60	1099 (20)	576 (24)	89 (29)	47 (18)	12 (12)	5 (4)	NA	-	-	11 (14)
		60 – <80	2446 (44)	1196 (50)	76 (24)	34 (13)	4 (4)	2 (2)	NA	-	-	18 (22)
		≥80	410 (7)	186 (8)	9 (3)	0	0	3 (2)	NA	-	-	17 (21)
Lazarevic et al., 2014 [42]	Sweden	Total	3251 (100)	810 (43)	NA	36 (1.9)	42 (2.2)	21 (1.1)	-	238 (13)	249 (13)	NA
		18–39	193 (6)	57 (37)	NA	11 (7.1)	16 (10)	3 (1.9)	-	7 (4.5)	11 (7.1)	NA
		40–59	612 (19)	212 (44)	NA	13 (2.7)	13 (2.7)	12 (2.5)	-	54 (11)	66 (14)	NA
		60–69	650 (20)	211 (45)	NA	6 (1.3)	8 (1.7)	0 (0)	-	56 (12)	62 (13)	NA
		70–79	1007 (31)	240 (41)	NA	4 (0.7)	5 (0.9)	6 (1)	-	91 (16)	86 (15)	NA
		80+	789 (26)	90 (45)	NA	2 (1)	0 (0)	0 (0)	-	30 (15)	24 (12)	NA
Enjeti et al., 2004 [26]	Singapore	Total	454 (100)	179 (39)	52 (11)	34 (7.5)	5 (1.1)	4 (0.9)	62 (13.6)	30 (6.6)	32 (7)	33 (7.3)
		15–35	102 (22)	35 (34)	17 (17)	18 (18)	1 (1)	1 (1)	7 (6.8)	3 (2.9)	4 (3.9)	6 (5.9)
		36–55	175 (39)	74 (42)	25 (14)	12 (6.9)	3 (1.7)	3 (1.7)	10 (5.7)	3 (1.7)	7 (4)	11 (6.3)
		56–75	148 (33)	60 (41)	9 (6.1)	4 (2.7)	1 (0.7)	nil	35 (23)	18 (12)	17 (11)	11 (7.4)
		>75	29 (6)	11 (38)	1 (3.4)	nil	nil	nil	10 (35)	6 (21)	4 (14)	5 (17)
Nakase et al., 2000 [16]		Total	350(100)	189 (54.0)	49 (14.0)	57 (16.3)	12 (3.4)	10 (2.9)	33 (9.4)			
	Japan	16–35	60 (17.1)	23 (28.4)	12 (14.8)	15 (18.5)	3 (3.7)	2 (6.2)	5 (6.2) *			
		36–55	134 (38.2)	69 (41.3)	21 (12.6)	24 (14.4)	8 (4.8)	5 (3.0)	7 (4.2) *			
		56–75	136 (38.9)	82 (49.1)	16 (9.6)	18 (10.8)	1 (0.6)	2 (1.2)	17 (10.2) *			
		76–95	20 (5.7)	15 (65.2)	0 (0)	0 (0)	0 (0)	1 (4.2)	4 (7.4) *			
		Total	192 (100)	102 (53.1)	27 (14.1)	11 (5.7)	14 (7.2)	6 (3.1)	31 (16.1)			
	Australia	16–35	58 (30.2)	30 (46.2)	6 (9.2)	7 (10.8)	7 (10.8)	3 (4.6)	5 (7.7) *			
		36–55	67 (34.8)	36 (45.6)	11 (13.9)	4 (5.1)	4 (5.1)	3 (3.8)	9 (11.4) *			
		56–75	57 (29.6)	32 (44.4)	8 (11.1)	1 (1.4)	3 (4.2)	0 (0)	13 (18.1) *			
		76–95	10 (5.2)	4 (28.6)	2 (14.3)	0 (0)	0 (0)	0 (0)	4 (28.6) *			
Cheng, Y. et al., 2009 [27]	China	Total	1290	547 (42)	187 (14)	109 (8)	NA	16 (1)	29 (2)	11 (1)	18 (1)	26 (2)
		0–9	13	2 (15)	3 (23)	2 (15)		1 (8)	0 (0)	0 (0)	0 (0)	0 (0)
		10–19	123	43 (35)	22 (18)	15 (12)		2 (2)	3 (2)	0 (0)	3 (2)	0 (0)
		20–29	186	69 (37)	29 (15)	19 (10)		4 (2)	2 (1)	0 (0)	2 (1)	4 (2)
		30–39	259	98 (38)	46 (17)	27 (10)		2 (1)	5 (2)	2 (1)	3 (1)	4 (2)
		40–49	273	118 (43)	46 (17)	20 (7)		2 (1)	8 (2)	4 (1)	4 (1)	7 (3)
		50–59	239	111 (46)	31 (13)	19 (8)		4 (2)	4 (2)	0 (0)	4 (2)	5 (2)
		60–69	141	76 (54)	5 (4)	7 (5)		1 (1)	4 (3)	3 (2)	1 (1)	2 (1)
		70–79	48	21 (44)	4 (8)	0 (0)		0 (0)	3 (6)	2 (4)	1 (2)	4 (8)
		80–89	8	8 (100)	0 (0)	0 (0)		0 (0)	0 (0)	0 (0)	0 (0)	0 (0)
Xin, L. et al., 2012 [12]	China	Total	2308 (100)	919 (39.8)	386 (16.7)	349 (15.1)	48 (2.1)	30 (1.3)	NA	NA	63 (2.7)	126 (5.5)
		0–9	156 (6.8)	50 (32.1)	22 (14.1)	35 (22.4)	3 (1.9)	9 (5.8)			1 (0.6)	1 (0.6)
		10–19	310 (13.4)	103 (33.2)	56 (18.1)	83 (26.8)	8 (2.6)	4 (1.3)			1 (0.3)	3 (1.0)
		20–29	371 (16.1)	150 (40.4)	77 (20.8)	53 (14.3)	12 (3.2)	4 (1.1)			3 (0.8)	5 (1.3)
		30–39	406 (17.6)	155 (38.2)	92 (22.7)	49 (12.1)	8 (2.0)	2 (0.5)			2 (0.5)	11 (2.7)
		40–49	405 (17.5)	162 (40.0)	77 (19.0)	57 (14.1)	10 (2.5)	4 (1.0)			3 (0.7)	14 (3.5)
		50–59	341 (14.8)	156 (45.7)	44 (12.9)	47 (13.8)	5 (1.5)	2 (0.6)			3 (0.9)	6 (1.8)
		60–69	200 (8.7)	93 (46.5)	12 (6.0)	13 (6.5)	1 (0.5)	4 (2.0)			2 (1.0)	8 (4.0)
		70–79	105 (4.5)	42 (40.0)	5 (4.8)	12 (11.4)	1 (1.0)	1 (1.0)			1 (1.0)	8 (7.6)
		80–89	14 (0.6)	7 (50.0)	1 (7.1)	0 (0.0)	0 (0.0)	0 (0.0)			1 (7.1)	0 (0.0)
Kadam, A.P.S. et al., 2016 [18] ^¥^	India	Total adult	1906 (100)	887 (46.5)	172 (9)	286 (15)	76 (4)	95 (5)	171 (9)	57 (3)	114 (6)	152 (8)
Boujmia O.K.A et al., 2021 [13]	Morroco	Total adult	789 (100)	347 (44)	31 (3.9)	66 (8.4)	37 (4.7)	NA	NA	NA	NA	42 (5.3)
Gmidène et al., 2012 [14]	Tunisia	Total	631 (100)	234 (37.1)	83 (13.2)	78 (12.2)	22 (3.5)	NA	33 (5.3)	14 (2.2)	19 (3)	44 (7)
		≤ 15	97 (15.3)	22 (3.5)	16 (2.6)	18 (2.8)	3 (0.5)	NA	6 (0.9)	2 (0.3)	4 (0.6)	8 (1.3)
		>15	534 (84.7)	212 (33.6)	67 (10.6)	60 (9.4)	19 (3)	NA	27 (4.3)	12 (1.9)	15 (2.4)	36 (5.7)
Shaikh, M.S. et al., 2018 [17]	Pakistan	Total adult	288 (100)	176 (61.1)	14 (4.9)	24 (8.3)	2 (0.7)	NA	na	na	3 (1)	7 (2.4)
Meng et al., 2013 [15]	Malaysia	Total	480 (100)	334	11	36	NA	NA	-	4	6	NA
		0–14	61 (12.7)	35 (10.5)	2 (18.2)	12 (33.4)	NA	NA	-	-	-	NA
		15–30	121 (25.2)	86 (25.7)	3 (27.2)	9 (25.0)	NA	NA	-	-	4 (66.6)	NA
		31–40	75 (15.6)	59 (17.7)	2 (18.2)	6 (16.7)	NA	NA	-	-	-	NA
		41–50	82 (17.1)	64 (19.2)	2 (18.2)	2 (5.5)	NA	NA	-	-	1 (16.7)	NA
		51–60	70 (14.6)	46 (13.8)	1 (9.1)	5 (13.9)	NA	NA	-	3 (75)	1 (16.7)	NA
This study 2021, Malaysia	Malaysia	Total	854 (100)	294 (100)	86 (100)	51 (100)	30 (100)	15 (100)	32 (100)	-	-	19 (100)
		13–15	24 (3)	3 (1)	2 (2)	5 (10)	1 (3)	0 (0)	0 (0)	-	-	2 (11)
		16–19	58 (7)	21 (7)	8 (9)	8 (16)	1 (3)	0 (0)	2 (6)	-	-	1 (5)
		20–29	124 (14)	46 (16)	20 (23)	12 (23)	8 (27)	5 (33)	0 (0)	-	-	1 (5)
		30–39	152 (18)	52 (18)	21 (24)	11 (21)	9 (30)	2 (13)	6 (19)	-	-	2 (11)
		40–49	137 (16)	48 (16)	17 (20)	6 (12)	4 (13)	3 (20)	2 (6)	-	-	1 (5)
		50–59	168 (20)	65 (22)	12 (14)	6 (12)	7 (23)	1 (7)	9 (28)	-	-	5 (26)
		60–69	128 (15)	49 (17)	5 (6)	1 (2)	0 (0)	4 (27)	9 (28)	-	-	6 (32)
		>70	63 (7)	10 (3)	1 (1)	2 (4)	0 (0)	0 (0)	4 (13)	-	-	1 (5)

Numbers (%); some (%) may not add up to 100 due to rounding. * Includes all abnormalities of 5/7/8. ^¥^ Number of cases were calculated from the percentages given. ^§^ This study included patients with secondary AML. NA: no data available.

## Data Availability

Data used in this study is available as Appendix A.

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
