# Peer review of "Genetic Profiles and Risk Stratification in Adult De Novo Acute Myeloid Leukaemia in Relation to Age, Gender, and Ethnicity: A Study from Malaysia"

_ijms, 2021, doi:10.3390/ijms23010258_

Round 1

Reviewer 1 Report

Very well performed study that validated inferior outcomes among older patients and implied inferior genomic evolution with age among those with AML.  I would simply ask that the authors qualify that in the era of targeted therapies and druggable mutations, including IDH, that the analysis may be limited and not comprehensive in the contemporary era to provide context.

Author Response

Thank you very much for the comments and suggestions. As the National Reference Centre for Haematology in Malaysia, there are only limited mutations panels performed on AML patients due to limited diagnosis and treatment options resources. We would like to make the best use of the data available in our setting to elucidate the genetics of AML in relation to age, gender and ethnicity, and with more resources and testing in future, we will be able to include more genetic markers and assess the benefits of targeted therapies and druggable mutations.

Reviewer 2 Report

Ambayya et al. evaluated genetic profiles, Moorman hierarchical classification, and 2017 ELN-based risk stratifications in relation to age, gender, and ethnicity in adult patients with AML.
Although the work is carefully done and described, I don't think the results are really having a significant impact. I believe the conclusions are not consistent enough for the journal's impact.

Author Response

Thank you very much for the comments and suggestions. In this study, as mentioned by the Reviewer, we have carefully evaluated and described the genetics profiles, Moorman hierarchical classification, and 2017 ELN-based risk stratifications in relation to age, gender, and ethnicity in adult patients with AML. We believe these findings are significant enough to provide the groundwork and an overview of the genetic landscape of adult AML in Malaysia, as one of the largest multi-ethnic countries in Southeast Asia.

Reviewer 3 Report

Ambayya and colleagues proposed an article entitled “Genetic profiles and risk stratification in adult de novo acute myeloid leukaemia in relation to age, gender, and ethnicity: a study from Malaysia”.

The article deals with the cytogenetic profiles of AML patients with NPM1 and FLT3-ITD mutations in the Malaysian population with three ethnic groups Malays, Chinese, and Indians. In 2013 Chen et al have reported the cytogenetic profile of AML patients in Malaysia with a cohort of 480 patients.

In the present study, the authors present a similar kind of data but the cohort size is significantly larger with 854 AML patients which provides the authors to derive more substantial and meaningful data.

The manuscript is straightforward, easy to understand, and within the scope of  MDPI-IJMS.

I would be okay with publishing this work in  MDPI-“IJMS” once the following minor points are addressed:

  • Throughout the paper, each citation is given in individual square brackets (only one citation 1 bracket). This practice is uncommon and not seen in international standard journals. I would suggest the authors to put multiple references in one square bracket with a comma separating them(if multiple citations are there).
  • Page2, line:4 the authors mention normal karyotype but they have to initially describe what they mean by normal karyotype for the easy understanding of the readers.

Author Response

Thank you very much for your comments and suggestion. Our response to first the comment: This bracket citation was based on the journal’s requirement to use the bracket for each citation. If the editor requires a change of this format, we will be happy to change the format using our citation tool (Mendeley).

2. We have rephrased the second sentence to enable better comprehension of the readers as follow:

Other chromosomal aberrations, including trisomies and normal karyotype, are categorised as an intermediate-risk group.[4]–[7]

Round 2

Reviewer 2 Report

Thanks for your reply, but I remain with the idea expressed above. The data is purely descriptive and I don't think it is meaningful enough.

Author Response

Dear Reviewer,

Thank you very much for your comments and feedback. We have attached the reply, kindly review.

Thank you. 
